# Oncolytic Herpes Simplex Virus Type 1 Induces Immunogenic Cell Death Resulting in Maturation of BDCA-1^+^ Myeloid Dendritic Cells

**DOI:** 10.3390/ijms23094865

**Published:** 2022-04-27

**Authors:** Philipp Kalus, Jolien De Munck, Sarah Vanbellingen, Laura Carreer, Thessa Laeremans, Katrijn Broos, Inès Dufait, Julia K. Schwarze, Ivan Van Riet, Bart Neyns, Karine Breckpot, Joeri L. Aerts

**Affiliations:** 1Laboratory for Neuro-Aging and Viro-Immunotherapy (NAVI), Vrije Universiteit Brussel (VUB), 1000 Brussels, Belgium; philipp.kalus@vub.be (P.K.); jolien.de.munck@vub.be (J.D.M.); sarah.vanbellingen@vub.be (S.V.); laura.carreer@vub.be (L.C.); thessa.laeremans@vub.be (T.L.); 2Laboratory for Molecular and Cellular Therapy (LMCT), Vrije Universiteit Brussel (VUB), 1000 Brussels, Belgium; katrijn.broos@vub.be (K.B.); karine.breckpot@vub.be (K.B.); 3Department of Radiotherapy, Laboratory of Translational Radiation Oncology, Supportive Care and Physics, Vrije Universiteit Brussel (VUB), 1000 Brussels, Belgium; ines.dufait@vub.be; 4Department of Medical Oncology, Universitair Ziekenhuis Brussel (UZ Brussel), 1000 Brussels, Belgium; juliakatharina.schwarze@uzbrussel.be (J.K.S.); bart.neyns@uzbrussel.be (B.N.); 5Stem Cell Laboratory, Departement of Hematology, Universitair Ziekenhuis Brussel (UZ Brussel), 1000 Brussels, Belgium; ivan.vanriet@uzbrussel.be

**Keywords:** oncolytic virus (OV), melanoma, immunogenic cell death (ICD), myeloid DCs (myDCs), cancer immunotherapy

## Abstract

Recently, a paradigm shift has been established for oncolytic viruses (OVs) as it was shown that the immune system plays an important role in the specific killing of tumor cells by OVs. OVs have the intrinsic capacity to provide the right signals to trigger anti-tumor immune responses, on the one hand by delivering virus-derived innate signals and on the other hand by inducing immunogenic cell death (ICD), which is accompanied by the release of various damage-associated molecules from infected tumor cells. Here, we determined the ICD-inducing capacity of Talimogene laherparepvec (T-VEC), a herpes simplex virus type 1 based OV, and benchmarked this to other previously described ICD (e.g., doxorubicin) and non-ICD inducing agents (cisplatin). Furthermore, we studied the capability of T-VEC to induce the maturation of human BDCA-1^+^ myeloid dendritic cells (myDCs). We found that T-VEC treatment exerts direct and indirect anti-tumor effects as it induces tumor cell death that coincides with the release of hallmark mediators of ICD, while simultaneously contributing to the maturation of BDCA-1^+^ myDCs. These results unequivocally cement OVs in the category of cancer immunotherapy.

## 1. Introduction

Regulated cell death (RCD), and especially apoptosis, has traditionally been considered a silent type of cell death, in contrast to accidental necrotic cell death. As research into RCD evolved from morphological observations towards the integration of biochemical and functional analysis, new types of RCD were discovered. At the same time, this advance led to new approaches of categorizing RCD, based on both morphological as well as functional features. Immunogenic cell death (ICD) represents a particular type of cell death that is capable of evoking both tumor-specific innate and adaptive immune responses under certain circumstances (e.g., chemotherapy, radiotherapy, oncolytic virotherapy, protease inhibitors), making it highly relevant for new anti-cancer therapeutics [1]. In contrast to other types of RCD, which are generally linked to certain cell death pathways, ICD is more a catchall classification based on functionality, which is demonstrated by the fact that several forms of RCD, including apoptosis, necroptosis and ferroptosis amongst others, can lead to ICD [2,3,4,5,6]. The concept of ICD first emerged in the context of chemotherapy, as researchers realized that tumor-specific immune responses determined the efficacy of traditional anti-cancer treatment [7]. Obeid et al., expanded the insight into ICD by demonstrating that the inoculation of CT26 tumor cells that have been treated with various agents (e.g., doxorubicin, staurosporine, etc.) into immunocompetent BALB/c mice conveyed protection against a subsequent rechallenge with untreated CT26 cells. They further identified that this protection strongly correlated with a translocation of calreticulin (CRT) from the endoplasmic reticulum to the cell surface [8,9]. Later, similar observations were noted in in vivo models where mice were treated with doxorubicin, which led to the fact that it is nowadays generally accepted as an ICD inducer [10,11,12].

CRT translocation is regarded as an indispensable mediator of ICD, alongside vesicular release of ATP and release of high-mobility group box 1 (HMGB-1) but also secretion of type I interferons, release of annexin A1 and tumor-associated nucleic acids [13]. Appreciating ICD as an important concept in cancer treatment and more specifically in immunotherapy, oncolytic viruses (OVs) represent another promising advance in the pursuit of new cancer treatments. The term OVs comprises viruses that selectively replicate in tumor cells and not (or to a far lesser extent) in their healthy counterparts [14]. In most cases, specific genetic modifications were made to enhance this tumor-specific replication, thus improving the safety and efficacy of OVs. Depending on the type of OV, these can involve deletions of viral genes to improve tumor selectivity and/or the insertion of exogenous genes, e.g., to enhance immunogenicity.

Talimogene laherparepvec (T-VEC) is a herpes simplex virus type 1-derived OV (oHSV-1) and so far, the only OV approved by both the Food and Drug Administration and European Medicines Agency for the treatment of local and advanced cutaneous melanoma [15,16,17,18]. Although investigations into the ICD inducing capacity of OV treatment remain scarce, there are studies showcasing the potential of OVs to release ICD mediators in melanoma or squamous cell carcinoma models [19,20,21]. 

Dendritic cells (DCs) play an important role in the stimulation of the adaptive immune response. Upon the induction of (immunogenic) cell death, cells release a variety of molecules into their environment depending on both cell type and cell death mechanism (CDM), which contribute to DC attraction. At the site of cell death, DCs engulf tumor-associated antigens, which in conjunction with other signals triggers their maturation and enables them to migrate to sentinel lymph nodes. In the lymph nodes, they present tumor antigens to naive T cells. Human conventional DCs type 2 (cDC2s), characterized by the expression of BDCA-1, are the most abundant type of myeloid DCs (myDCs) in the blood circulation. These cells have been shown to be able to reinvigorate the cancer immunity cycle and are key to the cross presentation of tumor antigens [22,23].

In this study, we treated melanoma cell lines with T-VEC and evaluated the effects on phosphatidylserine (PS) exposure, CRT translocation and ATP release in response to T-VEC treatment. To better estimate the scope of effects, we benchmarked those results to four different chemical cell death inducing (CDI) agents: cisplatin (CDDP), doxorubicin (DX), staurosporine (STS) and sulfazalasine (SFZ). CDDP is a chemotherapeutic agent previously described as a cell death inducer without ICD characteristics [24,25]. DX is an established anti-tumor drug described to induce ICD [8,10]. STS is known to induce apoptosis and presumably ICD [8,26,27], whereas SFZ is less studied in the context of cell death compared to the aforementioned agents, but is postulated to be an inducer of ferroptosis in cancer cells [28].

Although T-VEC treatment led only to minor levels of cell death and low exposure of PS, we found substantial CRT translocation and ATP release that was comparable to or even exceeded levels obtained when treating melanoma cell lines with chemical CDI agents. We also found that T-VEC induces maturation of BDCA-1^+^ myDCs. Our results show that oHSV-1, in particular T-VEC, are potent inducers of ICD, thus opening the perspective to further improve their treatment potential by exploiting this property.

## 2. Results

### 2.1. Screening Method to Identify Appropriate Concentrations of Cell Death Inducing Agents

When setting up experiments in the context of cell death, an important prerequisite is establishing the concentrations that cause an appropriate amount of cell death for each cell line within a certain timeframe. Reference concentrations can be found in the literature for many agents, but these are generally only available for commonly used cell lines. Moreover, because of extensive differences in susceptibility between different cell types, even from the same histology, fine tuning to determine the appropriate concentrations for subsequent experiments is almost invariably required. Therefore, in a first experiment, we treated various human melanoma cell lines (624-mel, 888-mel, 938-mel, CHL-1 and MZ2) with DX, STS and SFZ and performed a real-time monitoring of cell growth dynamics using the IncuCyte^®^ live analysis system (Appendix A). A selection of cell lines was also treated with CDDP and T-VEC. While DX affected all cell lines treated, we found SFZ and STS treatment affecting cells to varying degrees. While MZ2, 938-mel and CHL-1 were highly sensitive, 624-mel and 888-mel were more resistant to these treatments. Based on the proliferation curves and microscopic pictures, we determined that the CHL-1 cell line displayed the best dose-response profile for each of the treatments. For this reason, we selected this cell line for further analysis using the ICD assays.

Since proliferation curves and microscopic images lack a proper quantification of cell death, we performed a follow up experiment to quantify the amount of cell death for each of the treatments. Thus, we treated CHL-1 cells with the above-mentioned agents at concentrations adjusted to the results from the previous experiment for 24 h or 48 h and assessed DAPI positivity via flow cytometry, as a measure for cell death (Appendix A). Based on these results, for each agent, we selected concentrations that resulted in 10–50% cell death.

### 2.2. Cytocidal Effects of T-VEC Manifest Slower than Chemical Agents

In order to obtain a more accurate picture of the dynamics of cell death induced by the various CDI agents, we performed Annexin V/DAPI staining. PS exposure on the outer cell membrane (Annexin V positive) in cells that still maintain membrane integrity (DAPI negative) is generally considered a hallmark of early apoptosis, whereas cells displaying Annexin V/DAPI double positivity represent either late apoptotic cells or cells that have undergone other types of regulated cell death such as necroptosis or ferroptosis. CHL-1 cells were treated with CDDP (25 and 50 µM), DX (2.5 and 5 µM), STS (25 and 50 nM) or SFZ (1 and 1.5 mM) for up to 48 h or with T-VEC (multiplicity of infection (MOI) 10 and MOI 20) for up to 72 h and viability (DAPI) was assessed; 624-mel cells were treated with T-VEC (MOI 10 and MOI 20).

We observed a significant reduction in the viability after 24 h (CDDP and STS) and 48 h (CDDP, DX, STS, SFZ) ranging from 19.5 to 33.0% (Figure 1a). For T-VEC treatment, we observed only a limited effect on cell viability. While after 48 h we only detected a significant reduction in viability in 624-mel cells, after 72 h a reduction was observed in both CHL-1 and 624-mel. The extent was distinctly lower though, compared to treatments with chemical CDI agents, ranging from 9.3 to 12.3% (CHL-1) and 9.7 to 15.7% (624-mel) (Figure 1b).

When we evaluated the exposure of PS (Annexin V positive) within the DAPI negative fraction, we found that chemical CDI agents elicited a much more pronounced effect on PS exposure than T-VEC treatment (Figure 1c,d and Appendix A), indicating that T-VEC does not induce significant levels of apoptosis. Although after 24 h only CDDP resulted in a significant increase of PS exposing cells (17.0%), 48 h post treatment, we saw substantial PS exposure following treatment with CDDP, DX and STS (28.3–74.5%). In clear contrast, neither SFZ nor T-VEC treatment led to significant levels of PS exposure compared to control conditions at any time point.

### 2.3. T-VEC Treatment Triggers Surface CRT Exposure at Comparable Magnitude as Chemical Agents

After investigating the cell viability, we sought to elucidate whether the applied treatments give rise to features of ICD. In a first instance, we assessed whether cells treated either with chemical CDI agents or T-VEC expose CRT, a protein that normally resides in the endoplasmaic reticulum but is externalized upon ICD induction, on their cell surface. To do so, we treated CHL-1 and 624-mel as described for the previous experiment and assessed surface CRT on DAPI negative cells (see Section 2.2; Appendix A). At 24 h, we saw an increase in surface CRT in DX (5 µM), STS (25 and 50 nM) and CDDP (25 µM) treatments. However, only for STS (50 nM) and DX (5 µM) this increase was statistically significant compared to the untreated controls (Figure 2a). When prolonging the treatment to 48 h, however, we found significantly elevated levels in CDDP (25 µM), DX (2.5 and 5 µM), STS (25 and 50 nM) and SFZ (1.5 mM) treatments ranging from 25.0 to 80.8%.

Although we did not observe significant levels of PS exposure upon T-VEC treatment, we did find that T-VEC treatment significantly affected CRT exposure in both CHL-1 and 624-mel (Figure 2b). Very intriguing though was the distinct kinetic between the two studied cell lines. While 624-mel already demonstrated significant CRT exposure after 24 h (24.0 and 30.3% for MOI 10 and 20 resp.), which further increased after 48 h (65.5 and 66.0%) and remained consistent at 72 h (71.0 and 68.7%), for CHL-1, it took 72 h to detect significant changes. After 72 h, however, T-VEC treatment resulted in a striking 65.0% and 64.7% CRT exposing cells (MOI 10 and 20 resp.).

### 2.4. Treatment with T-VEC and Chemical Agents Causes Release of Vesicular ATP

In addition to the exposure of CRT, we examined the effects on ATP release, a second hallmark of ICD. CHL-1 and 624-mel were treated as described for previous experiments (see Section 2.2; Appendix A). After 24 h, we observed a significant decrease in ATP release in CDDP (25 µM) and STS (25 and 50 nM) treated cells (32.0–49.0%) (Figure 3a). While at 48 h, these levels remained rather stable for CDDP and STS (33.8–53.2%), we now also detected a reduction in ATP levels within DX treated cells, which was marginally significant when treated with 2.5 µM DX (12.3%) and significant when treated with 5 µM DX (19.3%). Lastly, we did not observe a significant reduction in ATP levels in response to SFZ treatment at any given time point, compared to control conditions.

While the kinetics of surface CRT exposure were widely differing between CHL-1 and 624-mel cells in response to T-VEC, the kinetics of ATP release were quite alike. Within the first 48 h, we found no significant decrease of ATP levels due to T-VEC treatment (Figure 3b). However, after 72 h, we found a significant drop of intracellular ATP in both cell lines. In 624-mel, we observed a trend when treated with MOI 10 (28%) and a significant effect when treated with MOI 20 (46.7%). In CHL-1 both MOI 10 and 20 caused an apparent drop of ATP levels in 43.0% and 49.0% of the cells respectively.

### 2.5. BDCA-1^+^ myDCs Mature and Upregulate PD-L1 after Exposure with T-VEC In Vitro

To investigate whether T-VEC treatment has a direct effect on BDCA-1^+^ myDC maturation, we co-cultured BDCA-1^+^ myDCs in the presence of T-VEC (MOI 10), either live or heat-inactivated. A trend towards increased expression of maturation-associated markers CD80, CD83 and CD86 was observed, whereas the expression of CD40 and HLA-ABC did not change compared to the untreated controls (Figure 4 and Appendix A). 

These results indicate that both an active OV and viral-associated molecular patterns derived from the inactivated virus can induce maturation of BDCA-1^+^ myDCs. We evaluated whether an environment where cancer cells are dying after treatment with T-VEC induces maturation of BDCA-1^+^ myDCs (Figure 5 and Appendix A). To do so, we first treated 624-mel and 938-mel, both human derived melanoma cell lines, with T-VEC (MOI 1). After 24 h and 48 h, we harvested the supernatant (SN) and cultured the isolated BDCA-1^+^ myDCs in this conditioned medium. An upregulation of CD80, CD83 and CD40 was observed, which reached statistical significance for CD80 and CD40 when BDCA-1^+^ myDCs were treated with SN of T-VEC treated 938-mel, while the expression of CD86 and HLA-ABC did not change before and after treatment. These results are in line with our previously published data on BDCA-3^+^ myDCs [29] and indicate that when cancer cells are dying, being accompanied by the secretion of different soluble factors may contribute to the maturation of BDCA-1^+^ myDCs. PD-L1 on BDCA-1^+^ myDCs is upregulated when exposed to active, heat-inactivated or conditioned medium (Figure 6 and Appendix A), indicating that T-VEC is able to upregulate this marker.

## 3. Discussion 

Our study demonstrates that T-VEC treatment is capable of triggering the exposure of CRT and the release of ATP with an efficacy comparable to or even exceeding that of established chemical CDI agents such as DX or STS. Intriguingly, T-VEC was able to do so, despite being markedly less cytocidal than the chemical agents we assessed in our study. Our results further indicate that T-VEC induced cell death might be meditated via a mechanism other than apoptosis. Moreover, we were able to show that T-VEC itself and SN from T-VEC treated melanoma cells provide stimuli for the maturation of human BDCA-1^+^ myDCs isolated from peripheral blood.

We sought to examine how the viability of CHL-1 and 624-mel is affected by T-VEC. Within CHL-1 cells, we took an additional step and compared the treatment with T-VEC with four chemical CDI agents: CDDP, DX, STS and SFZ. 

We found that CHL-1 displayed a sensitivity to all of the above-mentioned agents. SFZ overall induced less pronounced cell death levels than CDDP, DX or STS. When CHL-1 and 624-mel cells were treated with T-VEC, we observed only moderate effects on viability for both cell lines, although clear inhibitory effects on cell proliferation can be observed using the IncuCyte^®^ Live-Cell Analysis System. In 624-mel, we started to detect a minor, but significant effect on viability after 48 h. After 72 h, we detected a subtle, but significant reduction in viability in both cell lines. Chemical agents exerted their cytocidal effect much faster than T-VEC did, which we attribute to viral kinetics that require more time. An additional aspect explaining our findings might be inherent to T-VEC itself, which possesses the infected cell protein 6 (*ICP6*) gene, a gene that has been described as an inhibitor of both apoptosis and necroptosis in human settings [30,31,32]. Delaying cell death for as long as possible is obviously an advantage for viral propagation as viruses by nature rely on the running machinery of their host cells. Our data highlight that treatment duration is a pivotal point to be considered within studies examining cell death induced by OVs. Comparing our work to that of other groups, we found high variance in the amount of cell death, but also time point of onset of cell death ranging from 12–120 h post infection [19,20,33]. Bommareddy et al. [19] looked at viability 5 days post infection and demonstrated that once a threshold MOI is reached, a further increase does not lead to more cell death within the same amount of time. We also did not detect significant differences in viability between MOI 10 and 20, which might indicate that we already reached the maximum effect for our particular cell lines and that the limiting denominator is treatment duration. Last but not least, it is important to stress that, amongst these articles, the methods for assessing viability are very diverse, which might be another reason explaining the observed variation.

Looking at PS exposure, we found that CDDP, DX and STS did cause significant PS exposure on the cell surface of melanoma cells. In all chemical CDI agents, levels were more elevated at 48 h. For CDDP and DX, we found that the concentration used had a strong impact on the level of PS exposure. T-VEC treatment did not convincingly induce PS exposure at any time in neither of the cell lines. Though we did detect low levels at 48 h and 72 h in 624-mel, these levels were not statistically significant. These observations are in line with previous ones by Takasu et al. [20], who also did not find PS single positive cells when treating murine squamous carcinoma cells (SSC) with RH-2, a *γ34.5* deleted HSV-1. Paradoxically, they also demonstrated that the addition of caspase inhibitor z-VAD reduced the amount of cell death in their experiments, which led them to the conclusion that RH-2 was causing apoptosis in SCC cells [20]. As PS exposure in most of the cases is linked to apoptosis [34,35], this might be indicative that cell death through oHSV-1 happens predominantly via a mechanism other than apoptosis. Inclusion of z-VAD in our experimental setup could further support or confound this theory. It is worth noting that there are reports about ectopic detection of PS in the context of necroptosis, ferroptosis and even non-canonical cell death pathways [36]. Clearly, there remain many unanswered questions about CDMs evoked by oHSV-1 as well as other OVs, which need to be explored in future studies.

Understanding the ICD marker kinetics of T-VEC may further provide vital information regarding its efficiency in cancer therapy. When evaluating CRT exposure, we made several intriguing observations. We were able to detect increasing levels of surface CRT in response to all chemical CDI agents. While SFZ caused a detectable increase after 48 h, for CDDP, DX and STS, we were already able to detect significant amounts of CRT 24 h post treatment. Levels substantially increased at 48 h for the latter three agents. Considering that CRT translocation is accepted as one of the hallmark mediators of ICD and the fact that CDDP is mostly described as a non-ICD inducing agent [24,25], we did not expect to detect significant levels of surface CRT after CDDP treatment.

Surprisingly, CDDP produced the highest percentage of CRT positive cells amongst all treatments, contradicting previous reports suggesting CDDP to be incapable of triggering CRT translocation. Our data sparks the idea that CDDP is not generally incapable, but that its capacity is much rather dependent on a cellular context. Having in mind the modest effect on cell viability and PS exposure, we were quite struck by the fact that T-VEC elicited high levels of CRT exposure for both cell lines, exceeding levels reached with well-studied ICD inducers DX and STS. Though a direct comparison between chemical agents and an oHSV-1 is not evident, we were left with the impression that with chemical agents the exposure of CRT is to some extent linked to the degree of cytotoxicity. This is reflected by the fact that lower treatment concentrations were paired with less cell death and fewer CRT positive cells. As for T-VEC, differences between used MOI were negligible, suggesting that effects were not so much dictated by the MOI but rather by the viral kinetics which differed tremendously between both cell lines.

We next examined vesicular ATP release and found CDDP, DX and STS to cause release in CHL-1 cells. While CDDP and DX are known triggers for ATP release in cancer cells [25,37], to our knowledge, this property has never been demonstrated for STS. It is worth noting that lower concentrations of STS were sufficient to trigger significant ATP release, yet they failed to significantly induce CRT exposure. We further showed that T-VEC induces ATP release in both CHL-1 and 624-mel cells. Other than CRT, kinetics barely differed between both cell lines. In both, significant ATP release was detected after 72 h. The amount of release after T-VEC treatment was comparable or even outperformed levels of chemical agents. 

The next steps in the cancer immunity cycle are: (1) the activation and maturation of immature DCs and (2) the activation of T cells and thereby the induction of anti-cancer immune responses. We studied the impact of T-VEC and SN from T-VEC treated melanoma cells on the maturation of BDCA1^+^ myDCs. We observed an increased expression of CD80, CD83 and PD-L1 when BDCA-1^+^ myDCs were exposed to T-VEC itself, either active or heat-inactivated. Next to that, we observed an upregulation of CD40 after we exposed BDCA-1^+^ myDCs to the SN of 624-mel cells treated with T-VEC. In contrast, we observed an upregulation of CD80, CD83, CD86, CD40 and PD-L1 after we exposed BDCA-1^+^ myDCs to the SN of 938-mel cells treated with T-VEC. Although we should include more replicates to obtain statistically significant results, these results are an indication that active OV and pathogen-associated and danger-associated molecular patterns may contribute to the maturation of BDCA-1^+^ myDCs. These findings are in line with previous studies. As shown by Tijtgat et al., BDCA-1^+^, as well as BDCA-3^+^ myDCs, are able to mature and engulf tumor fragments of T-VEC treated melanoma cells and subsequently cross-present tumor antigens toward antigen-specific T cells [29]. Concerning PD-L1, the question arises whether the upregulation on myDCs might in fact represent a hinderance to downstream T cell activation. Nowadays, the combination of PD-1/PD-L1 blocking antibodies with oncolytic viruses is actively investigated as reviewed by Malogolovkin et al. [38]. In mouse vaccination studies using single domain antibodies against PD-L1, an increase in T cell activation was seen [39]. Despite these reports, a definite answer to this question is still missing, exemplified by work from Gogas et al. Although the results in a phase I clinical trial were very promising [40], in a follow up phase III trial (NCT02263508), the combination of T-VEC and pembrolizumab showed no improved progression free survival in patients with advanced unresectable stage IIIB-IVM1c melanoma [41]. 

Findings in this study sculpt the idea that oHSV-1 have the potential to be formidable agents in cancer treatment, since infected cells expose/release several DAMPs, while host cells are still viable, allowing viral propagation. This idea is supported by previous studies that also demonstrated OVs to have immunogenic properties. Bommareddy et al., studied ICD in SK-MEL-28 treated with T-VEC and reported HMGB-1 release, CRT exposure and ATP release [19,42]. Recently, Ma et al., characterized virus-mediated ICD by Adenoviruses, Semliki Forest virus and Vaccinia virus. They showed that all viruses mediate oncolysis, leading to the release of DAMPs, triggering the maturation of DCs, which leads to downstream activation of antigen-specific T cells [43]. Yamano et al., demonstrated HMGB-1 and ATP release in murine colon carcinoma cells after treatment with an oncolytic adenovirus and that vaccination with virus treated cells can protect from a follow up challenge with untreated tumor cells [44].

In conclusion, we found that T-VEC treatment exerts direct and indirect anti-tumor effects as it induces tumor cell death, coinciding with the release of hallmark mediators of ICD, while simultaneously contributing to myDC maturation. These results conclusively cement OVs in the category of cancer immunotherapies. They further feed the idea that a combination therapy consisting of T-VEC and intratumoral injection of conventional DCs type 1 (cDC1s) might have a synergistic effect, an approach which we are currently investigating in an in vivo mouse melanoma model. Similarly, an ongoing phase I clinical trial (NCT03747744) is conducted in which autologous cDC1 (BDCA-1^+^) myDCs and T-VEC are both injected intratumorally.

In the narrative of cell death, it becomes increasingly clear that the immunogenicity of cell death is influenced by the particular CDM that is induced but, concerning oHSV-1, CDMs remain poorly understood. In subsequent studies, we attempt to unravel underlying CDMs caused by oHSV-1 treatment. Deeper knowledge on this topic might help us to better understand the circumstances during cell death, allowing us to manipulate and take advantage of its immunogenic nature. Ultimately, this might help to improve existing treatments by exploiting cell death to their advantage or lead to novel or combined treatment strategies.

## 4. Materials and Methods

### 4.1. Cells and Cell Lines 

Human melanoma cell lines 624-mel, 888-mel, 938-mel (kindly provided by prof. S. Topalian, Institute for Cancer Immunotherapy, Johns Hopkins University School of Medicine, Baltimore, Maryland, USA), CHL-1 and MZ2 (ATCC, Manassas, VA, USA) were cultured in RPMI 1640 (Biowest, Nuaillé, France; L0501), supplemented with 10% FBS (Tico Europe, Amstelveen, Netherlands; FBSEU500), 2 mM L-glutamine (Sigma-Aldrich, Saint-Louis, MO, USA; G7513), 100 U/mL penicillin (Sigma-Aldrich, Saint-Louis, MO, USA, P0781) and 100 µg/mL streptomycin (further referred to as cRPMI) at 37 °C and 5% CO_2_. 

Isolated BDCA-1^+^ myDCs were obtained from patients included in different clinical trials (NCT03747744, NCT03707808) which have been conducted by the Department of Medical Oncology at the UZ Brussels. Those clinical trials were conducted in accordance with the Declaration of Helsinki and were approved by the ethics committee of the UZ Brussels. The patients provided written informed consent to acknowledge use of residual cells for research purposes. In short, patients underwent a leukapheresis whereafter CD14^+^ and CD19^+^ cells were depleted and BDCA-1^+^ myDCs were isolated (Miltenyi, Leiden, Netherlands; CliniMACS platform) [45]. BDCA-1^+^ myDCs were cultured in X-VIVO-15 (Lonza, Basel, Switserland; BE02-060F) supplemented with 1% human serum albumin (CLS Behring, Mechelen, Belgium; Alburex20), 2 mM L-glutamine, 100 U/mL penicillin, 100 µg/mL streptomycin, 1 mM sodium pyruvate (Thermo Fisher Scientific, Brussels, Belgium; 11360070) and non-essential amino-acids (Sigma-Aldrich, Saint-Louis, MO, USA, M7145) at 37 °C and 5% CO_2_.

### 4.2. Induction of (Immunogenic) Cell Death

For the induction of cell death, 624-mel, 888-mel, 938-mel, CHL-1 and MZ2 cells were treated with either STS (6.25–125 nM) (Sigma-Aldrich, SS0044), DX (0.25–2 µM) (Sigma-Aldrich, D1515), SFZ (0.2–1.5 mM) (Sigma-Aldrich S0883) or CDDP (25–125 µM) (Sellekchem, Houston, TX, USA; NSC 119875) for up to 48 h. When treated with T-VEC (Imlygic^®^, Amgen, Machelen, Belgium), CHL-1 and 624-mel were exposed up to 72 h to an MOI of 10 or 20. As a positive control for cell death in general, cells were treated with 10% EtOH and were incubated for 30 min. To exclude cytotoxic effects caused by the dissolvent dimethyl sulfoxide (DMSO), we included a control condition matching the highest DMSO concentration reached due to the addition of a CDI agent. As a negative control, cells were left untreated.

### 4.3. Monitoring Cell Proliferation in Real Time Using IncuCyte^®^ Live-Cell Analysis System

The 624-mel, 888-mel, 938-mel, CHL-1 and MZ2 cells were seeded in cRPMI in a flat bottom 96-well plate at the indicated densities. After overnight incubation at 37° C, 5% CO_2_, cells were treated with different inducers of (immunogenic) cell death at indicated concentrations. Cells were monitored using an IncuCyte^®^ Live-Cell Analysis System (Essen Bioscience, Göttingen, Germany) for 72 h with phase contrast image acquisition of 2 pictures per well every 2 h. Cell surface confluency was analyzed with IncuCyte ZOOM software (Essen Bioscience, GUI Version 2018A). Data were plotted in confluency over the course of time.

### 4.4. Quantification of Percentage Cell Death Using Flow Cytometry

CHL-1 cells were seeded in cRPMI at a density of 70,000 cells per well in a 12-well plate. After overnight incubation, cells were treated with different cell death inducers at indicated concentrations. The percentage of FBS in the culture medium for this assay was lowered to 2.5%. After 24 h, 48 h or 72 h, viable, dying and dead cells were harvested. After washing the cells with PBS (VWR, Radnor, PA, USA; 092-0434), single cells were resuspended in 50 µL DAPI solution (0.03 µg/mL in FACS buffer (PBS-BSA-azide); Sigma-Aldrich, D9542). After an incubation of 10 min at 4 °C and in the dark, 250 µL FACS buffer was added and cells were immediately analyzed using flow cytometry (BD LRS Fortessa, BD Biosciences, Erembodemgem, Belgium). Flow cytometric data were analyzed with FlowLogic software (Miltenyi Biotec, Version 7.3).

### 4.5. Phosphatidylserine Flip-Flop to Outer Membrane Following Induction of Cell Death

CHL-1 or 624-mel cells were seeded in cRPMI at a density of 70,000 cells per well in a 12-well plate. After overnight incubation, cells were treated with different cell death inducers at indicated concentrations. The percentage of FBS in the culture medium for this assay was lowered to 2.5%. After 24 h, 48 h or 72 h viable, dying and dead cells were harvested. After washing the cells with PBS, single cells were resuspended in 100 µL composed of 2 µL AnnexinV-APC (1/50; BioLegend, Amsterdam, Netherlands; 640920), 3.3 µL DAPI solution (0.03 µg/mL) and 94.7 µL 1X Annexin V Binding Buffer (BD Biosciences, 55645). Cells were incubated for 20 min at 4 °C and in the dark. After incubation, 300 µL of 1X Annexin V Binding Buffer was added and cells were immediately analyzed using flow cytometry (BD LRS Fortessa). Flow cytometric data were analyzed with FlowLogic software (Miltenyi Biotec, Version 7.3).

### 4.6. Calreticulin Exposure Following Induction of Cell Death

CHL-1 or 624-mel cells were seeded at a density of 70,000 cells per well in a 12-well plate. After overnight incubation, cells were treated with different cell death inducers at indicated concentrations. The percentage of FBS in the culture medium for this assay was lowered to 2.5%. After 24 h, 48 h or 72 h, viable, dying and dead cells were harvested. After washing the cells in PBS, single cells were resuspended in 50 µL composed of 0.5 µL anti-calreticulin-AF700 (1/100; Abcam, Cambridge, UK; ab196195) antibody or rabbit Ig G isotype control (1/100; Abcam, ab199093) and 49.5 µL FACS buffer. Cells were incubated for 30 min at 4 °C and in the dark. Afterwards, cells were washed in FACS buffer and resuspended in 50 µL DAPI solution (0.03 µg/mL). After an incubation of 10 min at 4 °C and in the dark, 250 µL FACS buffer was added and cells were immediately analyzed using flow cytometry (BD LRS Fortessa). Flow cytometric data were analyzed with FlowLogic software (Miltenyi Biotec, Version 7.3).

### 4.7. ATP Release Following Induction of Cell Death

CHL-1 or 624-mel cells were seeded at a density of 70,000 cells per well in a 12-well plate. After overnight incubation, cells were treated with different cell death inducers at indicated concentrations. The percentage of FBS in the culture medium for this assay was lowered to 2.5%. After 24, 48 h or 72 h, viable, dying and death cells were harvested. After washing the cells in PBS, single cells were stained in 50 µL of a 1/4000 dilution of Fixable Viability Dye eFluor780 (ThermoFisher Scientific, 65-0865-1B) in PBS for 20 min at room temperature in the dark. At the end of the incubation time, cells were washed with FACS buffer. Afterwards, cells were resuspended in 400 µL Krebs–Ringer solution (125 mM NaCl, 5 mM KCl, 1 mM MgSO4, 0.7 mM K_2_HPO_4_, 6 mM glucose and 2 mM CaCl2, 25 mM HEPES, pH 7.4) and were incubated for 5 min at room temperature. Cells were pelleted and resuspended in 300 µL quinacrine (0.125 µM; Sigma-Aldrich, 69-05-6; diluted in Krebs–Ringer solution) and incubated for 30 min at 37 °C and in the dark. After incubation, cells were washed and resuspended in FACS buffer. Cells were immediately analyzed using flow cytometry (BD LRS Fortessa). Flow cytometric data were analyzed with FlowLogic software (Miltenyi Biotec, Version 7.3).

### 4.8. Evaluation of BDCA-1^+^ myDC Maturation in Response to T-VEC Treatment

Upon treatment, BDCA-1^+^ myDCs were seeded in a 96-well round bottom plate at a density of 150,000–200,000 cells per well. BDCA-1^+^ myDCs were treated with either active T-VEC (MOI 10), heat-inactivated T-VEC (MOI 10; heat-inactivation: 15 min 65 °C, 1 min 100 °C) or conditioned medium, i.e., SN of dying 624-mel and 938-mel 24 h and 48 h after treatment with T-VEC (MOI 1). Untreated BDCA-1^+^ myDCs served as a negative control. As a positive control, BDCA-1^+^ myDCs were treated with a mix containing ssRNA fragments derived from HIV-1 long terminal repeat and protamine sulfate (PS/LTR). After overnight incubation, cells were harvested and stained with anti-CD11c-AlexaFluor 700 (BD Biosciences, 561352), anti-CD1c-BV510 (BD Biosciences, 742747), anti-CD80-PerCP-eFluor710 (ThermoFisher Scientific, 46-0809-42), anti-CD86-BV421 (BD Biosciences, 562432), anti-CD40-APC (BioLegend, 334310), anti-CD83-PE (BD Biosciences, 556855), anti-CD274-PE-CF594 (BD Biosciences, 563742) and anti-HLA-ABC-FITC (BD Biosciences, 557348) for 20 min at 4 °C and in the dark. Cells were acquired on the flow cytometer (BD LRS Fortessa) and data were analyzed with with FlowLogic software (Miltenyi Biotec, Version 7.3).

### 4.9. Statistics

Statistical analyses were performed with GraphPad Prism 8.3.1 software (Dotmatics, Stortford, UK). Results are expressed as mean ± standard deviation (SD). Data from the different cell death assays included at least three independent experiments and were analyzed using a two-way ANOVA or mixed model with repeated measurements followed by a Šidák correction. Phenotypic differences within the BDCA-1^+^ myDCs were analyzed using a Kruskal–Wallis test, followed by a Dunn’s multiple comparisons test to compare datasets and conditions. Three or four data points per condition derived from five independent patient samples are shown. The number of asterisks in the figures indicates the statistical significance as follows: * *p* < 0.05; ** *p* < 0.01, *** *p* < 0.001 and **** *p* < 0.0001.

## Figures and Tables

**Figure 1 ijms-23-04865-f001:**
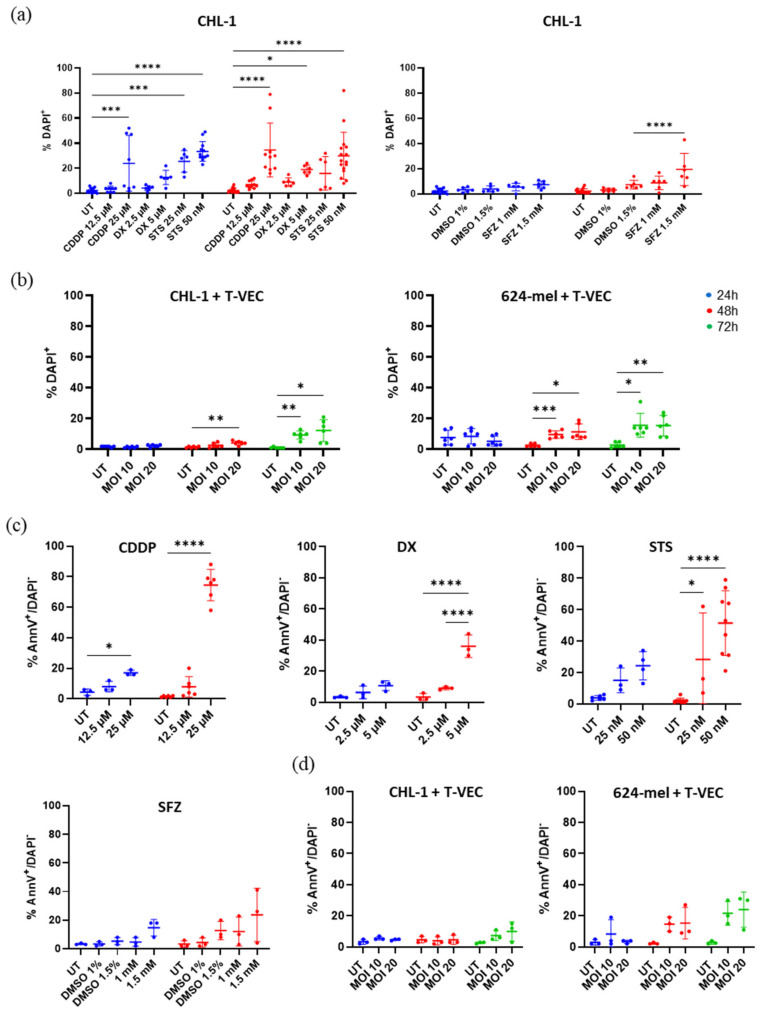
Cell viability and exposure of PS in response to various treatments. (**a**,**b**) CHL-1 cells were treated with CDDP, DX, STS, SFZ or T-VEC for up to 72 h. The amount of dead cells (DAPI positive) was determined by means of flow cytometry. (**c**) CHL-1 cells were treated with CDDP, DX, STS or SFZ for up to 48 h. (**d**) CHL-1 or 624-mel cells were treated with T-VEC for up to 72 h. Exposure of PS within the DAPI negative population was determined by means of flow cytometry. Data displays at least three repeats of independent experiments. Data were analyzed with two-way ANOVA or mixed model with Šidák corrections. * *p* < 0.05; ** *p* < 0.01, *** *p* < 0.001 and **** *p* < 0.0001 Abbreviations: CDDP: cisplatin; DMSO: dimethyl sulfoxide; DX: doxorubicin; MOI: multiplicity of infection; PS: phosphatidyl serine; SFZ: sulfasalazine; STS: staurosporine; T-VEC: Talimogene laherparepvec; UT: untreated.

**Figure 2 ijms-23-04865-f002:**
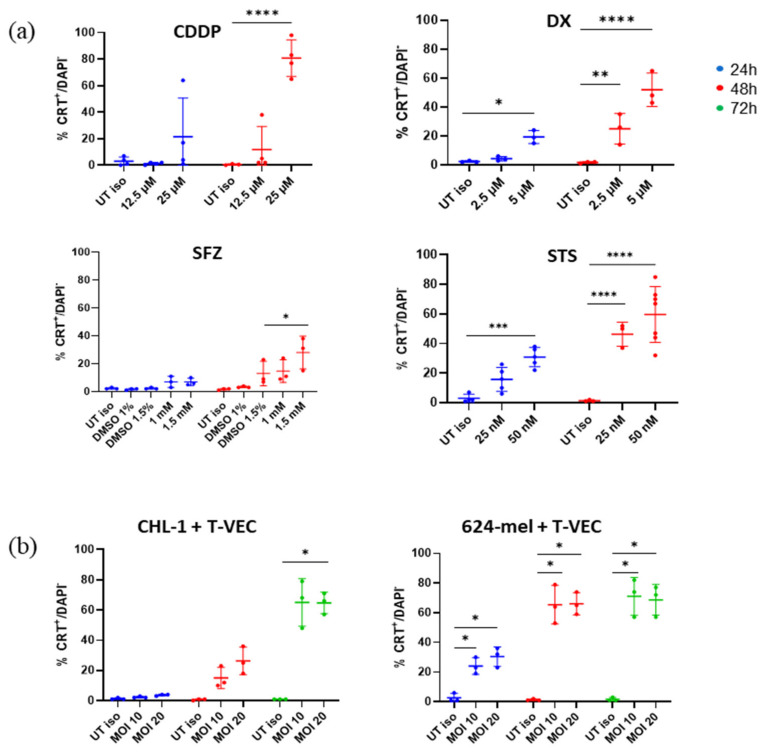
Exposure of surface CRT in response to various treatments. (**a**) CHL-1 cells were treated with CDDP, DX, STS or SFZ for up to 48 h. Within the DAPI negative population surface CRT was detected by means of flow cytometry. CRT levels were compared to untreated, isotype-stained cells. (**b**) CHL-1 or 624-mel cells were treated with T-VEC for up to 72 h. Within the DAPI negative population surface CRT was detected by means of flow cytometry. Data display at least three repeats of independent experiments. Data were analyzed with two-way ANOVA or mixed model with Šidák corrections. * *p* < 0.05; ** *p* < 0.01, *** *p* < 0.001 and **** *p* < 0.0001 Abbreviations: CDDP: cisplatin; CRT: calreticulin; DX: doxorubicin; iso: isotype control; MOI: multiplicity of infection; SFZ: sulfasalazine; STS: staurosporine; T-VEC: Talimogene laherparepvec; UT: untreated.

**Figure 3 ijms-23-04865-f003:**
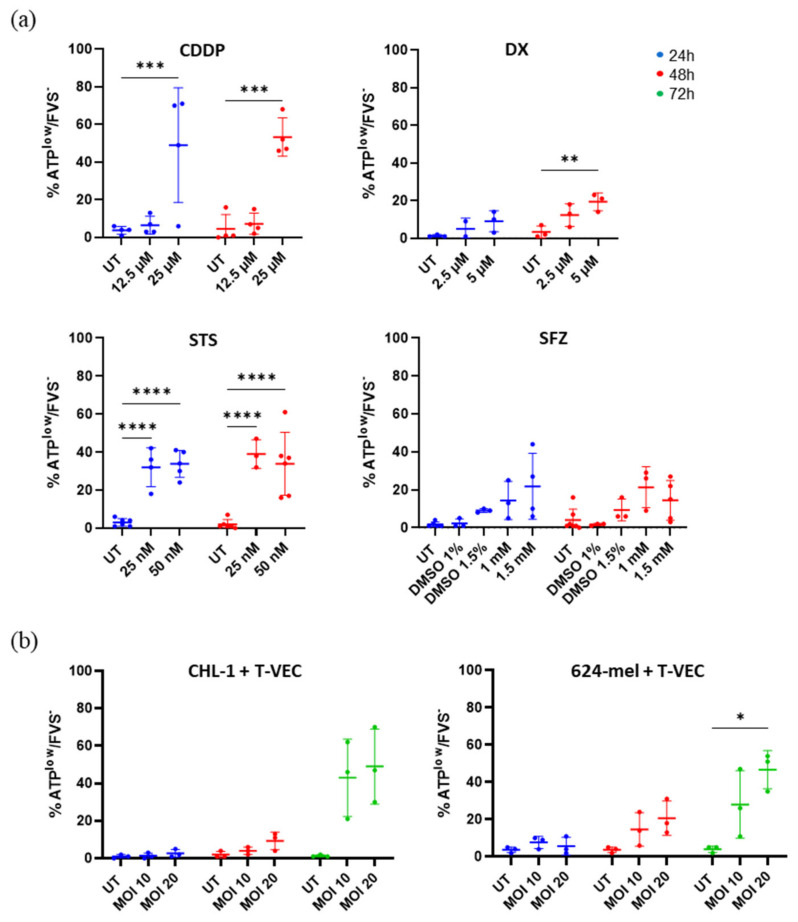
Release of vesicular ATP in response to various treatments. (**a**) CHL-1 cells were treated with CDDP, DX, STS or SFZ for up to 48 h. Within the FVS780 negative population, cells with reduced intracellular ATP (decreased quinacrine signal) were detected by means of flow cytometry. (**b**) CHL-1 or 624-mel cells were treated with T-VEC for up to 72 h. Within FVS780 negative population, cells with reduced intracellular ATP were detected by means of flow cytometry. Data display at least three repeats of independent experiments. Data were analyzed with two-way ANOVA or mixed model with Šidák corrections. * *p* < 0.05; ** *p* < 0.01, *** *p* < 0.001 and **** *p* < 0.0001 Abbreviations: CDDP: cisplatin; DX: doxorubicin; FVS: fixable viability stain; MOI: multiplicity of infection; SFZ: sulfasalazine; STS: staurosporine; T-VEC: Talimogene laherparepvec; UT: untreated.

**Figure 4 ijms-23-04865-f004:**
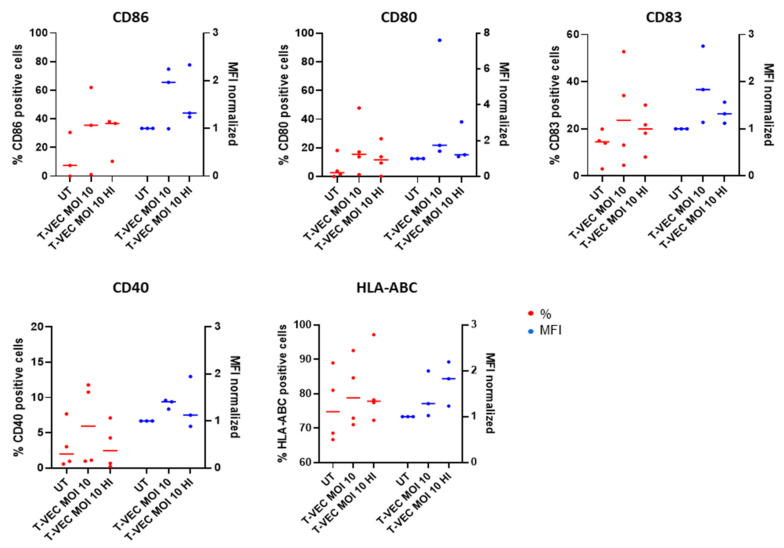
Maturation of BDCA-1^+^ myDCs after exposure with (in)activated T-VEC. Graphs show the expression of CD86, CD80, CD83, CD40 and MHC-I molecules (HLA-ABC) on BDCA-1^+^ myDCs after treatment with active or heat-inactivated T-VEC. As a negative control, cells were left untreated. Graphs display the percentages (red) and (normalized) MFI (blue) as median of different patient samples of at least three independent experiments. Data were analyzed using a Kruskal–Wallis test followed by Dunn’s multiple comparisons test. Abbreviations: myDCs: myeloid dendritic cells; HI: heat inactivated; MFI: mean fluorescence intensity; T-VEC: Talimogene laherparepvec; UT: untreated.

**Figure 5 ijms-23-04865-f005:**
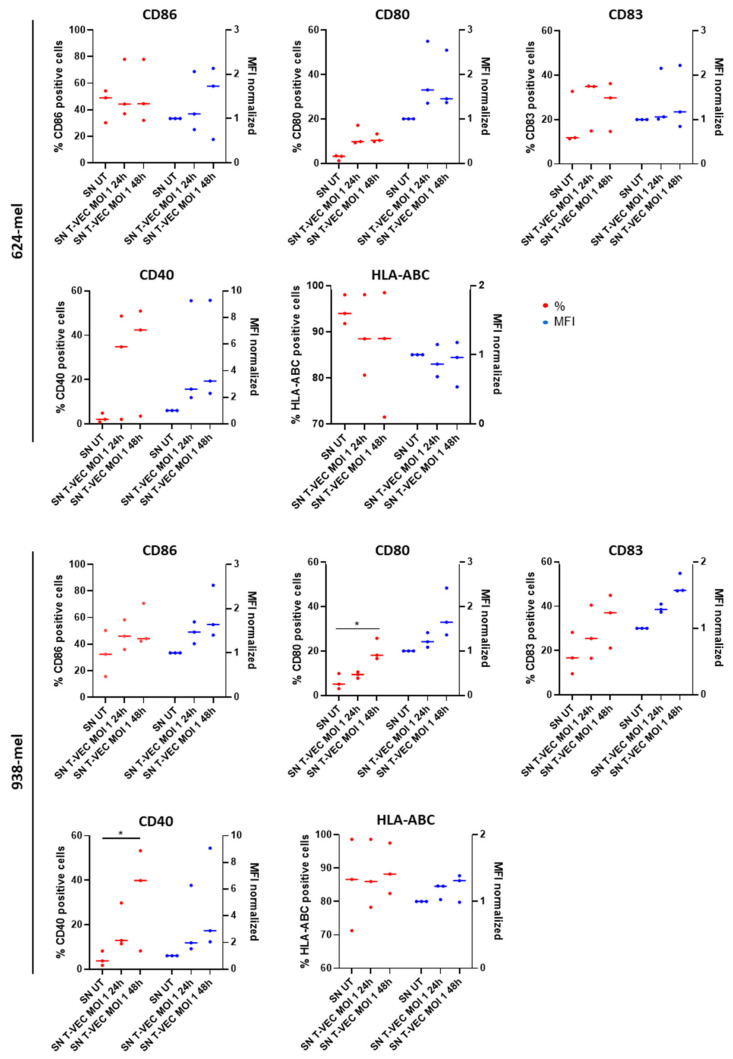
Maturation of BDCA-1^+^ myDCs after exposure to melanoma cells treated with T-VEC or conditioned medium. Graphs show the expression of CD86, CD80, CD83, CD40 and MHC-I molecules (HLA-ABC) on BDCA-1^+^ myDCs after treatment with conditioned medium (=SN of cells treated with T-VEC for 24 or 48 h. As a negative control, BDCA-1^+^ myDCs were exposed to SN of untreated cells. Graphs display the percentages (red) and (normalized) MFI (blue) as median of different patient samples of at least three independent experiments. Data were analyzed using a Kruskal–Wallis test followed by Dunn’s multiple comparisons test. * *p* < 0.05 Abbreviations: myDCs: myeloid dendritic cells; MFI: mean fluorescence intensity; SN: supernatant; T-VEC: Talimogene laherparepvec; UT: untreated.

**Figure 6 ijms-23-04865-f006:**
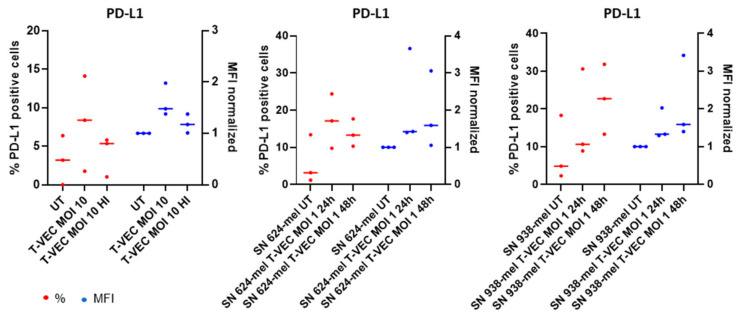
PD-L1 expression on BDCA-1^+^ myDCs after exposure with T-VEC or conditioned medium. Graphs show the expression of PD-L1 on BDCA-1^+^ myDCs after treatment with active T-VEC, heat-inactivated T-VEC or conditioned medium (SN of cells treated with T-VEC for 24 or 48 h). As a negative control, BDCA-1^+^ myDCs were left untreated or were exposed to the SN of untreated cells. Graphs display the percentages (red) and (normalized) MFI (blue) as median of different patient samples of at least three independent experiments. Data were analyzed using a Kruskal–Wallis test followed by Dunn’s multiple comparisons test. Abbreviations: HI: heat inactivated; myDCs: myeloid dendritic cells; MFI: mean fluorescence intensity; PD-L1: Programmed death-ligand 1; SN: supernatant; T-VEC: Talimogene laherparepvec; UT: untreated.

## Data Availability

The raw data supporting the conclusions of this article will be made available by the authors upon request.

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
