# Peer review of "Oncolytic Herpes Simplex Virus Type 1 Induces Immunogenic Cell Death Resulting in Maturation of BDCA-1^+^ Myeloid Dendritic Cells"

_ijms, 2022, doi:10.3390/ijms23094865_

Round 1
Reviewer 1 Report
The submitted manuscript is a very interesting example of the article taking into consideration the anticancer activity of oncolytic viruses. The introduction part including a description of immunogenic cell death is well prepared.
Material and methods are well and properly described. Statistical tests were chosen correctly.
The Results are properly described.
The Discussion section is prepared in a thoughtful way and a sufficient number of articles was cited.
Summarizing, I recommend this manuscript for publication when the Authors correct the abovementioned issues in the article.
Author Response
A point-by-point response to the reviewer’s comments was uploaded below.

Reviewer 2 Report
The authors analyze the presence of Oncolytic Herpes Simplex Virus 1 induces immunogenic cell death resulting in maturation of BDCA-1+ myeloid dendritic cells. Knowing the mechanisms of metabolic regulation of cells is of vital importance to control cancer. But it is necessary to highlight some modifications that authors must make before publishing their work:
Regarding the Introduction section, the authors have not made reference because they have focused on CDDP, SFZ, doxorubicin (DX) and staurosporine (STS); I am referring to the importance of these compounds in their study and how they affect the cellular mechanism to justify their subsequent methodology. Also indicate that Of the first two it would remain to say what the abbreviation corresponds to and they did not mention it in this section.
In the results section there is a reference to a supplementary figure (line 100, 111, 156, 212, 216, 226), but I have not been able to observe it because those figures do not exist
And with respect to the figures, the symbols “*, **, ****” appear, but the authors have not reflected what they refer to.
In discussion, the authors relied on what was obtained in results and in previous results. And finally, a section of conclusions is missing.
Regarding the methodology section, I think it has been described correctly so that other researchers can use it.
Author Response

(The authors gave the same response as above.)

Round 2
Reviewer 2 Report
Once the second version of the article "Oncolytic Herpes Simplex Virus 1 induces immunogenic cell death resulting in maturation of BDCA-1+ myeloid dendritic cells" has been reviewed, I think the article can be published.